# Evaluation of the Optimal Uses of Five Genotypes of *Musa textilis* Fiber Grown in the Tropical Region

**DOI:** 10.3390/polym14091772

**Published:** 2022-04-27

**Authors:** Juan Carlos Valverde, Mónica Araya, Dagoberto Arias-Aguilar, Charlyn Masís, Freddy Muñoz

**Affiliations:** 1Facultad de Ciencias Forestales, Universidad de Concepción, Victoria 500, Concepción 4030000, Región del Bío-Bío, Chile; 2Escuela de Ingeniería Forestal, Tecnológico de Costa Rica, Cartago 30101, Costa Rica; monicka1914@gmail.com (M.A.); darias@tec.ac.cr (D.A.-A.); charlynmasis@outlook.com (C.M.); fmunos@tec.ac.cr (F.M.)

**Keywords:** physical properties, mechanical properties, energetic properties, chemical properties, Costa Rica

## Abstract

Knowing the genotypes of *Musa textilis* and its fiber production properties is key for developing cultivars with homogeneous properties and focusing on specific products or market segments that generate added value to the fiber. For this reason, the objective was to determine the optimal use of five genotypes of *M. textilis* (MT01, MT03, MT07, MT11, and CF01) with high productivity grown in the tropical region of Costa Rica. Therefore, anatomical, physical-mechanical, chemical, and energetic analyses were carried out on these fibers to define whether any genotype has the ideal conditions for a specific use. The results showed differences between the genotypes, obtaining significant differences in physical-mechanical properties (tension, water retention, and color), chemical properties (holocellulose, lignin, extractives, and elemental values of nitrogen, carbon, and sulfur), and energetic properties (volatiles, ash, and caloric value thermogravimetric analyses), which resulted in the establishment of two groups of genotypes with a dissimilarity degree of 35%. The first group, composed of MT03 and MT01, presented characteristics suitable for paper production, biodegradable materials, and composite materials. On the other hand, the second group, made up of MT07, MT11, and CF01, showed properties suitable for textiles, heavy-duty fibers, and bioenergy.

## 1. Introduction

The development of policies to reduce greenhouse gases and the Industrial Revolution 4.0 have led to a change in the use of raw materials in various industries worldwide. [1,2,3]. A new generation of biomaterials has been established to optimize energy and water resources, reduce the use of chemical compounds, and minimize residues [4,5]. Natural fibers stand out among the raw materials with greater adaptability to new trends, being biodegradable materials, and renewable, with a wide versatility of use and a minimum production cost and impact on the environment [6,7]. It can be combined with synthetic materials to create resistant biomaterials that are applicable in the textile, construction, furniture, automotive, aviation, biomedical, and microelectronics industries [8,9]. Among the most used fibers are the following: *Ananas comosus* (pineapple), *Musa* × *paradisiaca* (banana), *Cocos nucifera* (Coconut), *Corchorus capsularis* (Jute), and *Musa texitilis* (Abacá) [10,11,12].

In the case of *M. textilis,* it is an herbaceous plant belonging to the Musaceae family, native to the Philippines [13,14], which grows in humid tropical climates with an optimum temperature of between 28 and 30 °C [15] and rainfall of more than 2000 mm per year [16], reaching heights of 6.5 m with diameters of 15 cm at the bottom [16,17]. The fiber is obtained from the pseudostem and is characterized by a high mechanical resistance (for intensive use), tolerance to salinity (for materials in contact with water or chemical solutions), and high percentages of cellulose and hemicellulose (for stationery and textiles). In addition, it shows the moisture stability and compatibility with thermoplastic or biodegradable polymers, enabling the development of smart materials [18,19]. It is also considered one of the most versatile fibers available and adaptable to multiple technologies and develops matrices with uniform fiber distribution [20,21].

However, the *M. textilis* fiber’s problem in the international market is its wide variability of mechanical, physical, chemical, and energetic properties [22]. Traditionally, plantations have focused on maximizing fiber production, reducing harvest time, and homogenizing the color and visual quality [21]. In addition, genetic improvement has focused on selecting genotypes with higher growth and production, lower nutritional consumption, resistance to pathogens, and adaptability to the site’s environmental conditions [23]. This approach has reduced the impact and development of new biomaterials that use the fiber of *M. textilis* due to lower production efficiency and the susceptibility of changing properties depending on the origin [24], aspects that do not occur with other species used for natural fibers [25]. The impact of property variability affects the restriction to markets with higher quality requirements (which would increase the profit margin), the use of fiber in highly technical and specialized industries (medical and electrical), and compliance with international quality and environmental management certifications [26,27].

An alternative to increasing the competitiveness of *M. textilis* is the identification and differentiation of fiber properties in each genotype [28]. Multiple studies developed in other fiber-producing species have shown the effect of the genotype in the industry; for example, Debnath et al. [29] determined improvements in paper quality by selecting *Musa* sp. genotypes with a low lignin content and high holocellulose content (cellulose + hemicellulose), which significantly reduced the use of chemicals in paper production. On the other hand, Diabor et al. [30] identified differences in the mechanical properties of the *Manihot esculenta* fibers, identifying genotypes with greater resistance and ideal use in structural products. For their part, Lundqvist et al. [31] with *Eucalyptus* spp. identified the ideal varieties for biomaterials resistant to fire and chemical exposure, allowing the substitution of highly polluting synthetic fibers. Finally, Rennebaum et al. [32] defined the fiber quality and optimal use of *Linum usitatissimum* as a reinforcement material for multiple industrial products and the importance of using genotypes with the ideal properties for each target market.

Over the next decade, the market for natural fibers will increase significantly, and production will need to adapt to the changes and be compatible with the principles of sustainability and efficient use of natural fibers [33]. In this scenario, the cultivation of genotypes of *M. textilis* with fiber properties compatible with a specific market is vital; it has to promote the development of new high-efficiency biomaterials and boost prices in the market, improving the income of producers and allowing buyers a higher quality product. Therefore, the objectives of this study were the following: i. characterize the physical, mechanical, chemical, and energetic properties of five genotypes of *M. textilis*; ii. differentiate genotypes according to their properties; iii. define the optimal use of each genotype according to the industry’s current demand. The study hypothesized that it is possible to differentiate and define the optimal use of each genotype according to the properties of its fiber, and based on this, fiber-reinforced composite studies can be developed for specific markets.

## 2. Materials and Methods

### 2.1. Genotypes and Study Site

The following five genotypes of *Musa textilis* were used: MT01, MT03, MT07, MT11, and CF01 (most used material in commercial production) from the genotype collection of the Agricultural Transfer Institute (INTA) in Costa Rica. This material was previously selected for having the highest levels of fiber production at the pilot plantation level. The material was collected on a plantation located in Guápiles, Limón, Costa Rica. At (10°15′ N, 83°46′ W), at an altitude of 825 m, an average annual temperature of 25 °C, and an annual rainfall of 4000 mm, with a rainy season from May to December and a dry season from January to April.

The site presented an inceptisol soil with a dominant composition of clay and silt; the site’s topography was characterized by being flat and with an optimal water infiltration system. At the chemical level, it showed a pH of 5.5 with the optimal nutritional conditions for developing the species (concentrations of nitrogen, phosphorus, and potassium) [17].

### 2.2. Fiber Processing

Five plants were selected at a flowering stage with average crop dimensions for each genotype. Pseudostem was defibrated using the Gölthenboth and Mühlbauer [17] technique with mechanical shredding and subsequently separated into first and second commercial quality fiber (differentiated by fiber color). The first quality fiber was used for the study, which was dried at 60 °C for 72 h until it reached a moisture content of less than 20%.

### 2.3. Anatomical Properties

For each genotype, 18 fibers of 10 mm in length were used and placed in a tabletop microscope model TM3000 scanning electron microscope (SEM) (Hitachi High-Tecnology Copr., Tokyo, Japan). Each fiber was photographed with a 300× and 400× optical magnification on the transverse side of the fiber. Next, the mean area, diameter, mean cell lumen thickness, and percentage of free space within the fibers were determined. The analysis was performed with Image J software version 2.44 (HNI, Bethesda, MD, USA) [34].

### 2.4. Physical-Mechanical Properties

Regarding physical properties, moisture content was evaluated in green conditions and after fiber drying, colorimetry, dry moisture content, water absorption, and density were analyzed. 

Five samples of 3 g of fiber from each genotype were used and placed in an Ohaus model MB 45 moisture analyzer (OHAUS, Newark, NJ, USA) to estimate green and dry moisture content. Colorimetry analysis was implemented with Valverde and Arias [35] methodology, using four 5 g per genotype samples placed uniformly in a press to measure the color with a standardized NIX Pro spectrophotometer CIE chromatography (Nix sensor Ltd., Hamilton, ON, Canada). The color was determined from the 400 to 700 nm range with a 10 mm diameter measuring port. The measurement of the specular component included (SCI mode) was taken at an angle of 10°, which is typical for the heterogeneous surface (D65/10), with a D65 (corresponding to daylight at 6500 K). The color was evaluated in CIELAB format, which generated the following three parameters to explain color consisting of: L* (lightness), a* (color trend from red to green), and b* (color trend from yellow to blue).

For fiber moisture retention, three samples of 2 g each were used per genotype and dried at 105 °C for 72 h. Subsequently, the samples were weighed and placed in containers with distilled water and weighed every 24 h for 240 h by ASTM D570-98 to determine the accumulated absorption in each period.

The mechanical analysis focused on the tensile test, where the ASTM D3822M-14 standard was used. For genotype, 30 fibers of 300 mm in length were applied, pressed into pieces of wood 30 mm long at their ends and installed in a Tinius Olsen H10 KT universal mechanical testing machine (Tinius Olsen TMC, Pasadena, CA, USA). The machine was programmed at a speed of 14 mm min^−1^. 

### 2.5. Chemical and Energetic Properties

For each genotype, three samples were used. The following chemical properties were determined: lignin content using the lignin test T222 om-02, holocellulose by Seifert [36], extractives in hot and cold water with ASTM D1110-84, extractives in sodium hydroxide with the test ASTM D1109-84, extractable with dichloromethane with ASTM D1108-84, extractives with ethanol-toluene with ASTM D1107-96, ash content with test ASTM D1102-84, volatile content with the test ASTM D1762-84, fixed carbon was evaluated with ASTM D3172-07a and caloric power with ASTM D5865-87. In addition, an Elementar model Vario Micro Cube analyzer (Elementar, Langenselbold, Germany) was implemented to estimate carbon (C), hydrogen (H), nitrogen (N), and sulfur (S).

Thermogravimetric analysis (TGA) was applied to three samples per genotype using the Sebio-Puñal methodology [37]. For this purpose, 7.5 g of fiber per sample (moisture content of 12%) were placed in an SDT model Q600 analyzer (TA Instruments, New Castle, DE, USA) with a gas flow rate of 100 mL min^−1^ and a calorimetric range of 20 to 800 °C with a heating ramp of 20 °C min^−1^ and a constant atmosphere of 100 mL min^−1^ of nitrogen.

### 2.6. Genotype Differentiation and Optimal Use

Differentiation between genotypes was performed with all the data obtained in the characterization, and multivariate analysis with the divisive method were used to identify the similarity between genotypes and determine the main properties that generate differentiation. Subsequently, each group of genotypes compared their average properties with the data reported by Simbaña et al. [19], del Río et al. [38], Narayana et al. [39], Muthu et al. [40], Girones et al. [41], Saragih et al. [42], Saragih et al. [43], and Reed et al. [44] for the production of different natural fiber products. As a result, the following three categories of use were established: high (it has optimal properties for use), medium (properties are compatible; however, they are not ideal), and minimal (poorly compatible properties).

### 2.7. Statistical Analysis

The study was developed using a mixed with lmne package v. 3.1-157 [45]. A variance analysis (ANOVA) was applied for each variable analyzed to determine significant differences between genotypes, if necessary, Tukey’s test was applied to identify genotypes with different behavior. Subsequently, a multivariate analysis was carried out with FactoMineR package v. 2.4 [46]. All analyses were performed in the R program version 4.2.1. at a significance of 0.05.

## 3. Results and Discussion

### 3.1. Anatomical Properties

Anatomical analyses did not identify significant differences between the five fiber genotypes (Figure 1). The fiber showed an average diameter of 1.52 mm; each microfiber showed an average diameter of 89.56 µm, a cross-sectional area of 6316.52 µm^2^, and an average cell wall thickness of 20.11 µm. A proportion of free percentage within 55.3% showed that the cultivars presented a remarkable homogeneity at an anatomical level.

The anatomical values were within the ranges reported by Simbaña et al. [21], Bautista [47], and Balakrishnan et al. [48], with a microfiber diameter from 70.33 to 110.45 µm, with a cross-sectional area that ranged from 4500 to 11,200 µm^2^, and an average wall thickness of 18.89 µm, slightly lower than that obtained in the study. Previous studies developed by Mukul [49] determined that the difference in anatomical properties of the genotypes is expressed when there are changes in the environmental conditions in which the plant develops (for example, thermal and hydric stress). When environmental conditions are homogeneous, anatomical differentiation tends to be minimal; therefore, it is recommended to evaluate the genotype under different environmental conditions and analyze the degree of anatomical variability.

### 3.2. Physical-Mechanical Properties

In physical properties (Table 1), no significant differences were obtained in fiber density, achieving an average value of 1.50 g cm^−3^. Regarding water absorption, differences were only found at 24 h, where MT03 and MT07 showed a lower absorption (an average of 83.50%) compared to the other genotypes (89.32%); this behavior varied at 48 and 72 h, where no differences in absorption were found (with average values of 94.39% and 99.06%). With fiber color in the green condition, the five genotypes showed similar values with a lightness (77.56 L*), and an average value of a* and b* (−2.62 and 1.56, respectively), which indicates that the fiber showed a slightly yellowish-green coloration with low lightness, resulting in low color saturation (average C* of 1.40). On the other hand, with fiber in dry condition, it was shown that the value of L* was reduced to 77.54 with increases in a* and b* (1.37 and 4.03, respectively), considering a yellow-reddish coloration of medium brightness but maintaining the tendency to show no differences in color between genotypes.

Fiber densities are within the ranges reported by Barba et al. [50]; the lack of density differentiation is due to the uniformity of the environmental conditions in which the genotypes developed. Therefore, as with anatomical analysis, tests should be performed with different environmental and nutritional conditions to analyze the variability of each genotype [51]. On the other hand, they report significantly lower values for water absorption at 48 h compared with the studies of Hirondo et al. [52], which determined a 10% higher absorption, a difference that can be associated with variations in cell wall that showed a 20% lower thickness. The water absorption capacity of the fiber is relevant when developing composite matrices or reinforcing materials and defining their industrial use; it is a characteristic related to hygroscopicity and fundamental in defining the target market for the fiber [53]. The genotypes with a low hygroscopicity allow the development of matrices with structural stability and a slow response to climate (specifically relative humidity), which allows their use outdoors and in parts that are in constant contact with water [54]. In contrast, the fibers with high hydroscope are ideal for developing materials that require plasticity and adaptability to the environment that are readily biodegradable and that in industrial processes use the least amount of water and energy (in the drying of the matrix) [55].

Regarding fiber color in green conditions, it was not possible to compare the results with previous studies because the fiber is marketed with a moisture content below 20%, and it is under this condition that color becomes vital in the sales process. In contrast, dry fiber showed a coloration different from Richter et al. [27] for *M. textilis*, with a decrease between 15 and 25% of a* and between 10 and 17% of b*, which denotes a more whitish fiber. These differences may be due to the post-drying period of the fiber, during which oxidation and degradation of extractives occur [56]. Traditionally, the color has been a determining variable when marketing and estimating its market value; fiber with a uniform whitish tendency shows a higher price due to the reduction of the chemicals and bleaching processes, which allows it to be used more efficiently in textiles, paper, or reinforcement of materials with colors previously defined by the market [40,52,55].

Concerning the mechanical characterization (Figure 2), the tensile test showed a variation in the maximum strength of the genotypes. Therefore, two groupings were obtained, the first made up of the genotypes MT01, MT03, and CF01, which presented a maximum tensile strength of 405 MPa with a maximum elongation of 7.9% and significantly lower values, compared to the second group made up of MT07 and MT11, which showed a maximum tensile strength of 615 MPa with a maximum elongation of 10.8%.

The tension values obtained in the genotypes with the lowest strength were found within the normal ranges (330–450 MPa) for *M. textilis* described by Gironès et al. [41] and Bande et al. [56]. Fibers with a tensile strength below 500 kPa are ideal for products or biomaterials requiring moderate strength and ease of recycling [57]. In contrast, MT07 and Mt11 can be considered high-strength genotypes ideal for matrices/products of intensive use, which must withstand temporary loads and have a long life cycle [58]. Vazquez et al. [3] highlight that *M. Textiles* fiber increases the resistance of biomaterials (with polymers) between 10 and 15%, with the advantage that it is more resistant than other natural fibers. Its percentage in the matrix can be reduced, which allows the mixture to be optimized and adapted according to the requirements of each industry.

### 3.3. Chemical Properties

Chemical characterization (Table 2) showed the following conditions: i. For holocellulose, only the MT03 genotype showed a significantly different value (87.91%) than the other genotypes, which showed no statistical differences (average 90.67%). ii. Lignin showed a variation according to the genotype; MT07 and MT11 showed high values (an average of 14.81%), in contrast to MT01, MT03, and CF01, which were lower than 11.95%. iii. MT07 and MT11 showed the lowest values in cold water, ethanol-toluene, and dichloromethane extractives; in the case of hot water extractives, only MT07 showed significant differences; iv. extractives with sodium hydroxide did not show differences between genotypes; the average value was 1.31%. An elemental analysis showed that MT07 and MT11 indicated statistical differences in C, H, N, and S composition; on the other hand, the remaining three genotypes showed no significant differences.

Fiber chemical differentiation is critical to optimizing genotype utilization; Moreno and Protacio [59] demonstrate that in paper and textile production, the high percentages of holocellulose are essential to generate higher quality products and reduce the use of chemicals for lignin degradation. Furthermore, the extractives are relevant for the reinforcement of materials and the creation of stable matrices because they affect the adaptability of the fiber to be combined with synthetic materials or other natural fibers [38]. Our study shows the formation of two groups with different percentages of extractives, which shows that the use of the fiber must have a different response depending on the biomaterial used. Moreno and Protacio [60] reported that fibers with a low amount of extractives simplify the transformation and adaptability to reinforce biomaterials for structural use; on the other hand, fibers with a high extractive value are ideal for the extraction of compounds and their use in industries that require specific extractives to generate stability in mixtures. 

Elemental composition is an essential variable in the degradation and generation of fiber composites generally used in the electronics industry [41]. The differences in nitrogen values between genotypes have affected the fiber transformation efficiency and the quality of the target compounds; therefore, they should be as uniform and adaptable as possible according to the technology—method to be developed [61].

### 3.4. Energetic Properties

The energetic characterization (Figure 3) showed a variation of results according to the variable analyzed. In volatile content (Figure 3a), the MT11 and CF01 genotypes showed significantly higher percentages (on average, 83.14%) than the other three fiber genotypes, which did not differ. On the other hand, with ash content (Figure 3b), the MT01 genotype showed the highest ash values (1.8%), followed by MT03 (1.4%), and a grouping formed by MT07, MT11, and CF01 with an average value of 0.9%. In contrast, the carbon fix (Figure 3c) showed a minimum value in MT11 and CF01 genotypes (<9%), and the other three genotypes showed values of >24%. Finally, with the net caloric value (NCV) (Figure 3d), the genotypes MT07 and MT11 showed the highest (an average of 18,380 kJ kg^−1^), followed by MT01 and MT03 with 17,290 kJ kg^−1^ and CF01 with the lowest caloric value, with only 16,800 kJ kg^−1^.

For calorimetric characteristics, no data were found for the volatile content of the species; for ash, the values obtained for the genotypes MT07, MT11, and CF01 were significantly lower than those reported by Jiménez et al. [62] of 1.3%. In contrast, the caloric power of MT07 and MT11 exceeded the values determined by Agung et al. [63] of 14,000 to 17,000 kJ kg^−1^. Differentiation in energetic properties facilitates the selection of genotypes according to their potential use as an energy source (low-quality fiber) or their use in biomaterials with different manufacturing processes and uses [64]. For example, fibers with a high tolerance to heat allow the generation of high-resistance matrices with high-temperature thermoforming techniques and reduce the use of chemicals to increase combustion resistance, reducing production costs [53,64]. Shamsuyeve et al. [65] mention that in a circular industry where waste is used as an energy source, the NCV and content of volatiles and fixed carbon are fundamental for energy transformation efficiency and the determination of the optimal combustion technology. When the NCV is higher, greater energy efficiency and greater energy autonomy can be generated, significantly reducing energy costs and waste management [66].

The TGA analysis (Figure 4) showed that in the first 200 °C of fiber degradation temperature, there were no differences between genotypes with an average weight loss of 18%. However, after 300 °C, the MT01 genotype showed the most significant loss of weight or mass, reaching 54% at 400 °C, at which cellulose and lignin are degraded. In contrast, the other four genotypes at the same temperature lost on average less than 40% of their weight, a behavior that was maintained up to 780 °C where the remaining weight of genotype MT01 remained at 38.9%, significantly higher than the other genotypes that presented an average remaining weight of 24.9%.

The performance of the MT01 genotype showed differences compared to the results reported by Aung et al. [63] and Mozón et al. [67]. In general, in the first 200 °C, degradation varied between 15 and 20%, a range shared by all the genotypes. After that, 300 °C fiber degradation increased, which led to differences between genotypes, with MT01 as the material with the lowest weight loss at maximum temperature (40%) and the other four genotypes with the most significant degradation, with losses greater than 70%. De la Rosa et al. [68] mentioned that this difference might be due to the percentages of cellulose, polymerization of waxes, and other fiber constituents.

By TGA analysis, Salvador et al. [69] report that for natural flax, cotton, hemp, and kenaf fibers, a critical degradation temperature is 320–350 °C. The results obtained with the fiber from this study follow those of Yang et al. [66], and indicate that natural lignocellulosic materials decompose thermally between 150–500 °C, particularly hemicelluloses, which degrade between 150–350 °C, cellulose between 275–350 °C, and lignin at a temperature between 250–500 °C.

### 3.5. Genotypes Differentiation and Optimal Use

Fiber characterization identified significant differences among the five genotypes, finding two groupings of similarity among the materials (Figure 5a). With a similarity of 89%, the first grouping consisted of materials MT01 and MT03, which maintained similarities in most of the properties analyzed. On the other hand, the second grouping consisted of FC01 and MT11, with a similarity of 79% and a similarity of 74%. A similarity of 67% was obtained among all the genotypes analyzed.

When analyzing the variability of the genotypes according to the properties analyzed (Figure 5b), it was found that the physical and anatomical properties did not show statistical differentiation; on the other hand, the chemical, energetic, and mechanical properties showed a tendency for two groupings, an aspect that influenced the degrees of similarity defined among the genotypes. 

Based on the properties of the identified fibers and the defined groupings, a proposal for the material’s potential use was developed (Table 3). The fiber was recommended for the MT01 and MT03 genotypes for stationery, biodegradable materials, and fast degrading composite materials. These products are proposed due to the need to have materials with high cellulose and hemicellulose content and low lignin content; the process of pulping and homogenization of the mixtures for stationery is facilitated, and the creation of higher quality paper is enhanced [59]. Ritcher et al. [27] mention the key to having low-resistance fibers to develop biodegradable biomaterials for single-use or short life cycles. Furthermore, Shibata et al. [70] recommended using fibers with low mechanical resistance and low lignin levels to use ecological, moderate-use, and low-value-added materials compatible with both genotypes.

On the other hand, MT07, MT11, and CF01 materials show ideal conditions for more intensive use with the possibility of being used in textiles, thermal resistance materials, or higher strength composite materials due to their high lignin content tensile strength and energy values [44]. Valasek et al. [58] and Simba et al. [21] recommend high-strength fibers for intensive use in materials exposed to outdoor, variable climatic conditions and a long-life cycle. For their part, Agung et al. [63] highlight that fibers with similar properties to the studied fibers are ideal for use in textiles due to their strength, low cellulose composition that reduces material degradation, and energy values that affect the strength of the material so that it is ideal for exposure to temperatures below 50 °C. Finally, Barba et al. [50] highlight the implementation of biocomposites with high-strength natural fibers, ideal for developing structures in automobiles or materials with moderate structural use and that are easy to recycle or reuse. According to Muñoz et al. [71], the adequate performance of a given natural fiber depends on its physical, thermal, and morphological properties, as these are some of the factors that can affect the excellent performance of lignocellulosic fibers in a composite material. This information is fundamental in exploring the properties of natural fibers as reinforcement materials in thermoplastic matrices and other applications.

Therefore, the results obtained confirm the proposed hypothesis. It is possible to differentiate and specialize *M. textilis* plantations oriented to specific markets/products and adapted to the fiber properties requested by the market. This type of result opens the possibility of developing and using biomaterials with high efficiency and adaptability to the industry [72]; resulting in an increase in product quality and a reduction in the use of water and chemicals in the process of adapting the fiber to each product, which generates short-term environmental and economic benefits [8]. Based on this result, it is possible to take the next step in the change of reinforcement polymers and to determine the optimal technology to produce products that reduce the consumption of resources and waste and that are competitive with the materials available on the market with a more significant environmental impact [73]. Differentiation and identification of genotypes should influence advances in creating new biomaterials and promote the transition to sustainable production systems [74].

## 4. Conclusions

The characterization of the five genotypes showed significant differences in fiber properties, which allowed the generation of two groupings. The first group formed by MT01 and MT03 stood out for having energetic, mechanical, and chemical properties that made the material show a high adaptability for use in stationery, materials of fast degradation, and composite materials of low intensity. Aspect deferred to the second group formed by MT07, MT11, and CF01, which showed optimal fiber characteristics for use in composite materials of high resistance, textiles, and materials with thermal exposure. Therefore, the differentiated use of genotypes could impact the availability of materials for specific use in a productive sector, which would impact the fiber that responds to market demand.

Therefore, it is critical to characterize the genotypes and initiate tests with plastics to determine the specific uses of biomaterials. Our results made it possible to begin research on the application of the different fibers in matrices with polymers and other natural fibers and evaluate the efficiency and quality of the products, which should influence changes in the use of the fiber and the creation of a new market. Therefore, these results are considered a first step to changing the paradigm of *M. textilis* plantations in the tropics.

## Figures and Tables

**Figure 1 polymers-14-01772-f001:**
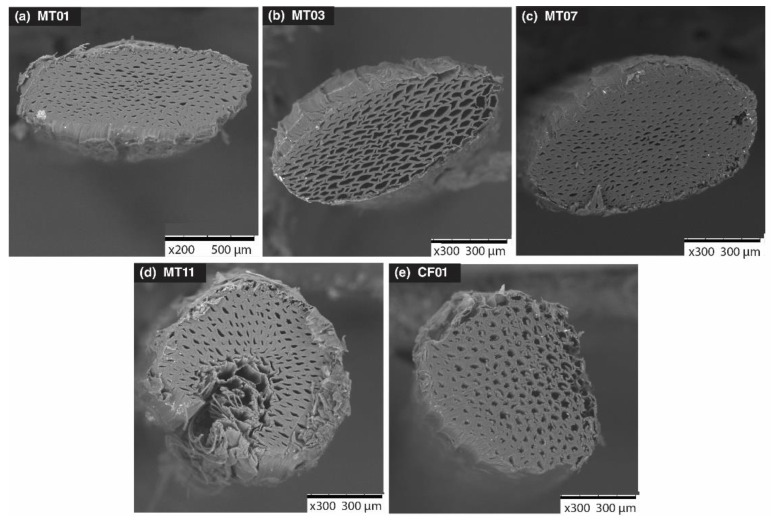
Electron microscopic cross-section view of the fibers of five genotypes of *M. textilis*.

**Figure 2 polymers-14-01772-f002:**
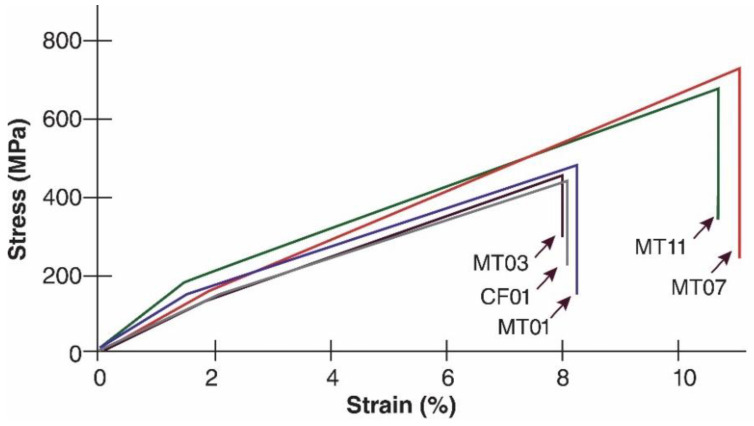
Fiber tension test for five genotypes of *M. textilis*.

**Figure 3 polymers-14-01772-f003:**
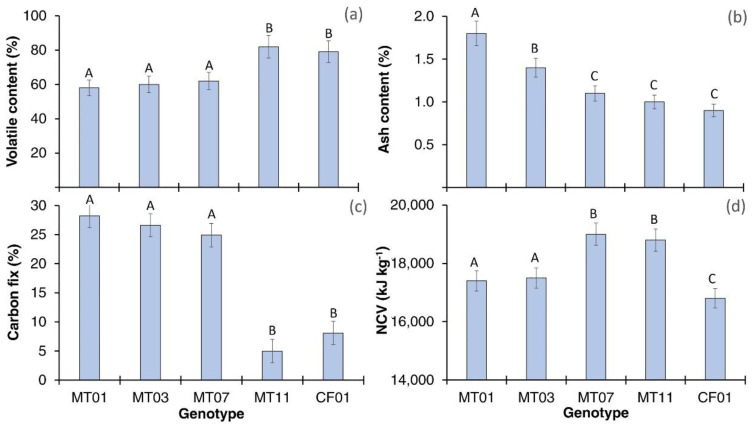
(**a**) volatile content; (**b**) ash content; (**c**) carbon fix; (**d**) NCV for five genotypes of *M. textilis*. *Note:* Different letters show significant differences at 0.05.

**Figure 4 polymers-14-01772-f004:**
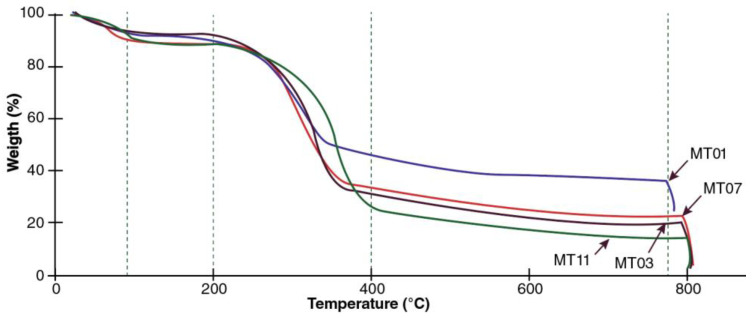
Fiber TGA test for five genotypes of *M. textilis*.

**Figure 5 polymers-14-01772-f005:**
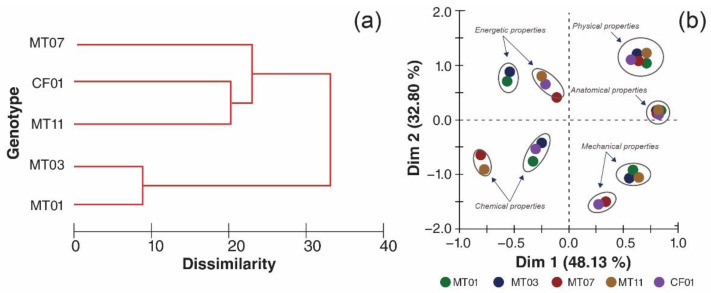
Cluster analysis (**a**); similarity clustering (**b**) of the fiber properties of five genotypes of *M. textilis*.

**Table 1 polymers-14-01772-t001:** Fiber physical properties of five genotypes of *M. textilis*.

Parameter		Genotype				
		MT01	MT03	MT07	MT11	CF01
Density (g cm^−3^)		1.52 A (0.02)	1.50 A (0.01)	1.49 A (0.02)	1.51 A (0.02)	1.50 A (0.03)
Green moisture content (%)		42.5 A (1.50)	45.80 A (3.33)	35.56 B (4.01)	41.22 A (4.21)	43.45 A (4.99)
Water absorption (%)	24 h	87.90 A (3.23)	82.90 B (2.89)	84.11 B (4.23)	90.11 A (3.78)	89.94 A (3.82)
	72 h	94.56 A (3.02)	93.38 A (2.11)	91.11 A (3.46)	95.99 A (2.87)	96.89 B (3.11)
	240 h	98.99 A (3.00)	99.02 A (2.34)	98.88 A (3.56)	99.50 A (3.17)	98.93 A (2.87)
Color (Green)	L*	89.90 A (2.22)	90.12 A (3.23)	90.25 A (3.55)	88.89 A (3.40)	90.98 A (3.45)
	a*	−0.09 A (1.33)	−2.89 B (3.09)	−0.90 A (2.67)	−0.34 A (0.23)	−2.90 B (3.02)
	b*	1.45 A (1.90)	2.89 B (2.88)	1.12 (2.08)	1.11 A (1.89)	1.08 A (1.22)
	C*	1.22 A (2.45)	2.55 B (2.39)	1.1 A (2.24)	0.10 A (2.88)	1.05 A (2.14)
Color (Dry)	L*	77.89 A (3.09)	74.89 B (2.80)	77.90 A (2.11)	78.80 A (3.45)	78.23 A (3.29)
	a*	1.18 A (1.22)	1.89 A (1.30)	1.44 A (1.24)	1.00 A (1.99)	1.34 A (2.04)
	b*	3.99 A (2.34)	4.50 A (1.33)	3.89 A (2.89)	3.99 A (2.09)	3.80 A (2.80)
	C*	3.56 A (2.33)	4.45 A (2.11)	4.30 A (2.90)	3.87 A (2.87)	4.11 A (2.33)

*Note:* Values in parentheses correspond to standard deviation; different letters show significant differences at 0.05.

**Table 2 polymers-14-01772-t002:** Fiber chemical properties for five genotypes of *M. textilis*.

Parameter		Genotype				
		MT01	MT03	MT07	MT11	CF01
Hollocelulose (%)		89.62 A (0.95)	87.91 B (0.46)	89.38 A (0.98)	93.06 A (0.37)	90.65 A (90.56)
Lignin (%)		11.43 A (0.28)	11.46 A (0.23)	15.49 B (0.39)	14.13 B (0.26)	11.66 B (0.10)
Extracts	Hot water (%)	11.01 A (0.18)	10.17 A (0.41)	7.75 B (0.24)	10.88 A (0.15)	3.90 C (0.20)
	Cool water (%)	11.27 A (0.24)	10.35 A (0.22)	6.76 B (0.20)	7.33 B (0.31)	3.67 C (0.16)
	Ethanol-toluene (%)	11.07 A (0.11)	10.43 A (0.23)	5.93 B (0.15)	7.19 B (0.16)	2.14 C (0.13)
	Sodium hidroxide (%)	1.33 A (0.10)	1.33 A (0.11)	1.30 A (0.09)	1.32 A (0.10)	1.31 A (0.09)
	Dichloromethane (%)	9.23 A (0.11)	9.44 A (0.09)	4.56 B (0.10)	5.01 B (0.09)	4.89 B (0.10)
Nitrogen (%)		0.14 A (0.02)	0.11 B (0.01)	0.09 B (0.02)	0.10 B (0.02)	0.10 B (0.02)
Carbon (%)		62.21 A (1.15)	66.44 B (0.74)	66.11 B (0.50)	66.49 B (0.45)	66.57 B (0.41)
Hydrogen (%)		6.44 A (0.02)	6.72 A (0.10)	6.61 A (0.12)	6.78 A (0.43)	6.90 A (0.29)
Sulfur (%)		1.55 A (0.09)	1.50 A (0.07)	1.32 B (0.03)	1.33 B (0.02)	0.33 B (0.03)

*Note:* Values in parentheses correspond to standard deviation; different letters show significant differences at 0.05.

**Table 3 polymers-14-01772-t003:** Potential uses for *M. textilis* fibers of each genotype.

Potential Use	Genotype					References
	MT01	MT03	MT07	MT11	CF01	
Paper	High	High	Medium	Medium	Medium	[38]
Materials with high degradation	High	High	Medium	Medium	Medium	[39]
Textile	Medium	Medium	High	High	High	[40]
Rope, heavy use	Medium	Medium	High	High	High	[41]
Composite materials (Low use)	High	High	Medium	Medium	Medium	[42]
Composite materials (High use)	Medium	Medium	High	High	High	[21,42]
Thermal exposition	Low	Medium	High	High	High	[43]
Energy	Low	Low	High	High	Medium	[44]
Electronic	High	High	High	High	High	[21]

## Data Availability

The data presented in this study are available on request from the corresponding author.

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
