# Peer review of "Evaluation of the Optimal Uses of Five Genotypes of Musa textilis Fiber Grown in the Tropical Region"

_polymers, 2022, doi:10.3390/polym14091772_

Round 1
Reviewer 1 Report
Musa textilis is an herbaceous plant belonging to the Musaceae family, native to the Philippines, which grows in humid tropical climates with an optimum temperature of between 28 and 30 °C and rainfall above 2000 mm per year, reaching heights of 6.5 m with diameters of 15 cm at the bottom. The use of M. textilis has focused on the generation of fiber from the pseudostem, characterized by its high mechanical resistance, so it is implemented in textiles, stationery, furniture, internal structures in automobiles and recently in microelectronics and nanocomposites. Knowing the genotypes of Musa textilis that have known fiber production properties is key for developing cultivars with homogeneous properties and focused on specific products or market segments that generate added value to the fiber. In this manuscript, characterization of the five genotypes showed significant differences in fiber properties, which allowed the generation of two groupings. The topic is important, the results are interesting and the methodology followed is appropriate, while the content falls well within the scope of this Journal. In general the paper makes fair impression and my recommendation is that it merits publication in this Journal, after the following major revision:
- The introduction should be reconstructed to present a coherent literature review. It may help the authors by answering the following questions: Why are these works relevant? Which specific problems were addressed? How are the previous results related with the latest work? What are the outstanding, unresolved, research issues? Answering the questions leads to the novelty of the proposed work naturally. I think this is essential to keep the interest of the reader.
- Materials and Methods part, Although the results look “making sense”, the current form reads like a simple lab report. The authors should dig deeper in the results by presenting some in-depth discussion.
- In Fig 4, the authors should give the explanations for the difference of data collected from different samples.
- The fiber composite materials has been widely used in the industry. The fiber composite materials is found to serve in many practical applications, such as fibrous porous materials, (see [Powder Technology, 2019, 349:92-98; International Journal of Heat and Mass Transfer, 2019, 137:365-371). Authors should introduce some related knowledge to readers. I think this is essential to keep the interest of the reader.
- Therefore, differentiated use of genotypes could impact the availability of materials for specific use in a productive sector, which would impact fiber that responds to market demand. Therefore, it is essential to highlight the importance of characterizing more genotypes and starting tests with plastics to determine the specific uses in biomaterials. The authors should give some explanation on above conclusions.
- Please expand the motivation, the problem context, clarify the problem description, and (if possible) add specific objectives.
- Please, expand the conclusions in relation to the specific goals and the future work.
- English grammar and syntax has to be checked carefully throughout the manuscript. There are several grammatical mistakes in the manuscript and it is very difficult to follow anything if they are not corrected.
Author Response
1. The introduction should be reconstructed to present a coherent literature review. It may help the authors by answering the following questions: Why are these works relevant? Which specific problems were addressed? How are the previous results related with the latest work? What are the outstanding, unresolved, research issues? Answering the questions leads to the novelty of the proposed work naturally. I think this is essential to keep the interest of the reader.
We appreciate the observation given, we consider that it is key to abide by it and have greater clarity in the introduction of the importance of knowing and differentiating the properties of the fiber, for which we made a restructuring and orientation of the entire introduction, which consists of L29 to L92
2. Materials and Methods part, Although the results look “making sense”, the current form reads like a simple lab report. The authors should dig deeper in the results by presenting some in-depth discussion.
The discussion section was expanded and especially the results in order to increase the depth of the work, therefore the results and discussion section added more analyzes and references that give greater clarity to the study, which is displayed from L178.
3. In Fig 4, the authors should give the explanations for the difference of data collected from different samples.
Corrected!
3. The fiber composite materials has been widely used in the industry. The fiber composite materials is found to serve in many practical applications, such as fibrous porous materials, (see [Powder Technology, 2019, 349:92-98; International Journal of Heat and Mass Transfer, 2019, 137:365-371). Authors should introduce some related knowledge to readers. I think this is essential to keep the interest of the reader.
Part of the restructuring of the article from the title, consider this observation in order to expand the impact and importance of the work.
4. Therefore, differentiated use of genotypes could impact the availability of materials for specific use in a productive sector, which would impact fiber that responds to market demand. Therefore, it is essential to highlight the importance of characterizing more genotypes and starting tests with plastics to determine the specific uses in biomaterials. The authors should give some.
The conclusions were improved and the observation was applied, the change is visible in L 410 explanation on above conclusions.
5. Please expand the motivation, the problem context, clarify the problem description, and (if possible) add specific objectives.
It is considered in the restructuring of the introduction and results.
6. Please, expand the conclusions in relation to the specific goals and the future work.
The conclusions were improved and the observation was applied,
7. English grammar and syntax has to be checked carefully throughout the manuscript. There are several grammatical mistakes in the manuscript and it is very difficult to follow anything if they are not corrected
Corrected!
Reviewer 2 Report
Comments for manuscript polymers-1649360-peer-review-v1
Manuscript: polymers-1649360-peer-review-v1
Title: Fiber characterization of five Musa textilis genotypes cultivated in the tropical region
In this paper, the authors used different types of Musa textilis genotypes cultivated in the
tropical region and analyze the anatomical, physical-mechanical, chemical and energetic
properties.
I think that the manuscript is original and is within the scope of the journal, but it is
necessary to address some points.
1. Lines 37-39: review articles cited there and also the wording. It is known that high
content of holcellulose (cellulose and hemicelluloses) is desired in a raw material
used to make paper pulp.
2. Line 141: include “and Discussion”
3. Lines 200-209 and Table 2: The result of extractives in ethanol-toluene is given there,
but these solvents are not mentioned in the Methodology chapter. On the other
hand, where are the results of extractives in soda and in dichloromethane? Include
these results.
4. Line 215: Correctly write the title of table 2.
5. Lines 124, 217 and Figure 3. I'm surprised why the authors didn't include the fixed
carbon result. This is part of the so-called proximal analysis (moisture, ash, volatile
material and fixed carbon) and is very important for the energetic characterization
of biomass. It is suggested that this output be included and will greatly help your
manuscript.
6. Line 259: It is suggested that the Methodology chapter include what is related to the
genotypes differentiation.
7. In some bibliographical references it does not appear in the name of the journal.
Please find the attached file.

Author Response
1. Lines 37-39: review articles cited there and also the wording. It is known that high content of holcellulose (cellulose and hemicelluloses) is desired in a raw material used to make paper pulp.
Corrected!
2. Line 141: include “and Discussion”
Corrected!
3. Lines 200-209 and Table 2: The result of extractives in ethanol-toluene is given there, but these solvents are not mentioned in the Methodology chapter. On the other hand, where are the results of extractives in soda and in dichloromethane? Include these results.
We appreciate the observation and offer our apology for the omission given with the extractives, in the correction the missing data has been added both in methodology and results and discussion.
4. Line 215: Correctly write the title of table 2.
Corrected!
5. Lines 124, 217 and Figure 3. I'm surprised why the authors didn't include the fixed carbon result. This is part of the so-called proximal analysis (moisture, ash, volatile material and fixed carbon) and is very important for the energetic characterization of biomass. It is suggested that this output be included and will greatly help your manuscript.
It is an important observation and by mistake the fixed carbon data was not added in the study, in its correction said graph was added, which is available in Figure 3c.
6. Line 259: It is suggested that the Methodology chapter include what is related to the genotypes differentiation.
We appreciate this observation, it was applied because it helps improve the quality of the article, the section was added in L161.
7. In some bibliographical references it does not appear in the name of the journal.
Corrected!
Round 2
Reviewer 1 Report
It is ok.
Author Response
On behalf of the authors, we thank you for your evaluation and for the observations that have inferred that the manuscript has a higher quality. Thank you
Reviewer 2 Report
Thank you very much for having heeded the suggestions made to your manuscript.
Please find attached file.

Author Response
On behalf of the authors, we thank you for your evaluation and for the observations that have inferred that the manuscript has a higher quality. Thank youLines 189-190: here the determination of extractives in sodium hydroxide (ASTM
D1109-84) is mentioned, but its results do not appear in Table 2.
We appreciate the observation, the missing data has been included in table 2.
2. Line 374: please write the title of table 2 correctly
Corrected!